

# Analysis of the Geomagnetic Component Z Daily Variation Amplitude Based on the China Geomagnetic Network

Xudong Zhao, Yufei He, Qi Li, and Xiaocan Liu

Institute of Geophysics, China Earthquake Administration, Beijing, 100081, China

*Correspondence to:* Yufei He (heyufei_bj@163.com)

**Abstract.** The daily variation amplitude of geomagnetic component Z is one of the important data products in Geomagnetic Network of China (GNC). It comes from the

difference between maximum and minimum of the component Z recorded by the geomagnetic instrument in a day. Based on this data product, the daily variation amplitude of Z is analyzed in the past twelve years (2008-2019), including variation for each month in high and low solar activity years, seasonal variations and comparisons between the stations in Yunnan Province and in southeast China. The study indicates that the ionospheric conductivity mainly contributes to the Z daily variations

amplitude in the same month or season changing along with solar activity. But the neutral wind in ionosphere could make the Z daily variations amplitude in equinox months equal to or greater than it in summer solstice months during some solar high activity years. Due to the complicated underground electrical structures in Yunnan province, the conductivity underground acts as an amplifier to make the Z daily variations amplitude increase by about 12% ~41% in Yunnan Province during equinox and summer

solstice months.

## 1 Introduction

The daily variation amplitude of geomagnetic component Z is calculated from the maximum minus the minimum of component Z in each day and it is an important data product of the Geomagnetic Network of China (GNC). The GNC completed the 'China Digital Seismic Observation Network' project in 2007

(Zhang et al., 2016). Until now, the GNC has more than 130 digital geomagnetic stations. And China becomes the country who has the most geomagnetic stations in the world. With the various quality control methods for geomagnetic data, the GNC could distribute the observatory data within 48 hours ( Xin and Zhang, 2011; Zhang and Yang, 2011).

Among the various quality control methods for geomagnetic data, the daily variation amplitude of

geomagnetic component Z is an effective data product to analyze the data quality for various instruments in the same station or to compare the data quality with different stations nearby.



The daily variation of geomagnetic field is mainly caused by Sq current, especially during the geomagnetic quiet time (Stening et al., 2005a, 2005b). The Sq current exists in the ionosphere and has a regular period of 24 hours (Orlando et al., 1993). In the middle and low latitude, the dayside current

system is mainly composed of two current vortices located on the north and south sides of the equator. The current vortex in the northern hemisphere flows clockwise, while the current vortex in the southern hemisphere flows clockwise (Richmond and Roble,1987). The center of the current vortex is in about 11:30 local time and ±30 °latitude (Xu, 2003). The feature of the Sq current makes the geomagnetic component Z recorded by stations of GNC decrease quickly near the noon every day. Based on this

phenomenon, the daily variation amplitude of Z component could represent the strength of Sq current and could also be used for different instruments or stations to compare each other to confirm if the data quality is up to standard.

Moreover, the daily variation amplitude of component Z could also be used to provide the normal background information to the anomaly extraction, such as earthquake. The variable magnetic field could

produce the induced magnetic field (Xu, 2009). Besides these two types of magnetic field sources, the horizontal component H mainly reflects the influence of outer space on the earth's magnetic field (such as storm), but the Z component also includes the influence of underground activities on the earth's magnetic field (Yuan et al.,2018). In seismic geomagnetism, it is generally believed that the geomagnetic vertical component Z is closely related to the underground medium and the daily variation abnormality before the

earthquake occurs (Feng et al., 2005). The spatial correlation method is a quantitative method to extract the diurnal variation distortion of geomagnetic vertical component Z (Lin and Shen, 1982). Other methods like load and unload response ratio also confirms the geomagnetic component Z diurnal variation amplitude significance in extracting the abnormally signal before earthquake (Zeng et al., 1996; Feng et al., 2000).

Stated thus, the daily variation amplitude of component Z is deserved attention. In this paper, we will not analyze the geomagnetic component Z daily variation amplitude relationship with earthquake. We will just intensive study the feature of component Z daily variation amplitude in China using the data after the completion of digital network until now on solar quiet days (2008-2019), including study the Z daily variation amplitude in different months, analysis the seasonal variation during more than ten years and

comparison the Z daily variation amplitude of Yunnan province (in southwest of China) with other regions. This study will provide the background information or reference to researchers who use the daily



variation amplitude of geomagnetic component Z to study the geomagnetic diurnal variation or to analyze the anomaly.

**2 Data and method**

The data used in this paper comes from all the stations of GNC. There are about 130 stations and over 150- instrument observation data during the past twelve years from 2008 to 2019 in this study. The distribution of stations is shown in Fig. 1. In GNC, the relative variation geomagnetic data comes from different instruments, mainly including the triaxial fluxgate magnetometer (e.g., FHDZ-M15, GM-4 and GM-3) and the proton vector magnetometer (e.g., FHD-1 and FHD-2). The triaxial fluxgate

magnetometer could provide the relative variation of component D, H and Z. The sampling rate is 1/s and the resolution is 0.1nT. The proton vector magnetometer could provide the geomagnetic total intensity F, horizontal component H and the component D. Its sampling rate is 1/min and the resolution is 0.1nT.Using the baseline of the proton vector magnetometer, the component Z could be calculated. The geomagnetic daily variation amplitude could be calculated by the

difference between the maximum and minimum through the data recorded by the geomagnetic instruments in a day. In Fig. 2, it is the Z component daily variation recorded by the triaxial fluxgate magnetometer GM-4 of Hongshan station (LYH, Latitude: 37.4 °N, Longitude: 114.7 °E) located in Hebei province near Beijing on March 18th, 2017. As shown in Fig. 2, due to that day is a solar quiet day, the Z component of geomagnetic station in north hemisphere has two extreme values located in the pre-noon

and afternoon in the daily variation, respectively. And the minimum value is near the noon. So the daily variation amplitude of Z component ($\Delta Z$) is the difference between the maximum value ($Z_{max}$) and the minimum value ($Z_{min}$) indicated by Eq. (1):

$$\Delta Z = Z_{max} - Z_{min} , \tag{1}$$

Before analysis of the $\Delta Z$, the data selection is as follow steps. This method is also used in the data

quality control of GNC. First, the larger error must be removed. As shown in Fig. 3 for example, the different color circles represent the data from different instruments. And the size of the circles means the value of the data. If the data appears evidently abnormal (e.g., the large green circle of station NAJ in Fig. 3), the inspection must be made to the instrument in that station and the invalid data is eliminated. Second, the data is selected through comparing the different instruments in the same station. If there are more than

two instruments in the same station, the key check is made to valid which instrument data is more effective corresponding to the different instruments error above 1 nT. Third, based on the two steps ahead, the select data in one station need to be compared with other stations nearby to make sure the data validity finally.



After making the validation of $\Delta Z$ in each day, the geomagnetic quiet days in every month is selected.

According to the list of international geomagnetic quiet days, for each station, the data from the quietest 5 days per month is averaged to represent the $\Delta Z$ value in each month, and expressed by $\Delta Zm$. $\Delta Zm$ is calculated from the Eq. (2). $\Delta Zi$ in Eq. (2) indicates the $\Delta Z$ for each quiet day in a month. The list of international geomagnetic quiet days is from the world geomagnetic data center, Kyoto, Japan (download site: http://wdc.kugi.kyoto-u.ac.jp/qddays/index.html). $\Delta Zm$ of every station for each month is

analyzed in the twelve years from 2008 to 2019 including the solar activity cycle 24.

$$\Delta Zm = \frac{\sum_{i=1}^{5} \Delta Z_i}{5}, \tag{2}$$

In order to study the seasonal variation of $\Delta Z$, the Lloyd's season is introduced. In geomagnetism, Lloyd's season is usually used to analyze the seasonal variation, that is, a year is divided into three seasons: March, April, September and October are the equinox months (spring equinox and autumn

equinox), expressed by E; May, June, July and August are the summer solstice months, expressed by J; and November, December, January, and February are the winter solstice months, expressed by D ( Xu 2003 ). The seasonal variation of $\Delta Z$ for each station could be calculated from the Eq. (3) and expressed by $\Delta Zs$. $\Delta Zm_i$ in Eq. (3) indicates the $\Delta Zm$ for each four months in every Lloyd's season.

$$\Delta Zs = \frac{\sum_{i=1}^{4} \Delta Zm_i}{4}, \tag{3}$$

Using the methods above, the daily variation amplitude of Z will be studied in different months, seasons and years. Also, it is used to inspect the difference of the geomagnetic diurnal variation in some local regions.

**3 The monthly variation $\Delta Zm$ in high and low solar activity year**

During the geomagnetic quiet days, the Sq current mainly causes the diurnal variation of geomagnetic

field. The generation of Sq current system is closely related to the sun, so the activity of the solar will directly affect the current system (Zhao, 2014). According to the frequency and average number of sunspots on the surface of the sun, the solar activity has a period of about 11 years (Usoskin, 2009). Fig. 4 indicates the sunspots variation from 2008 to 2019. The data distribution here is the monthly mean total sunspot number. It is obtained by taking a simple arithmetic mean of the daily total sunspot number over

all days of each calendar month. And the sunspot number data is from Sunspot Index and Long-term Solar Observations (SILSO data/image, Royal Observatory of Belgium, Brussels). The sunspot number



shows that the period from 2010 to 2016 was high solar activity time, and the peak appeared in 2014. The solar minimum occurred in December 2019, marking the start of a new solar cycle announced by experts from NASA and NOAA

(https://www.nasa.gov/press-release/solar-cycle-25-is-here-nasa-noaa-scientists-explain-what-that-mea ns/). For comparison, the monthly variation ΔZm for one year is analyzed in 2014 (high solar activity) and 2019 (low solar activity).

Using the data from all the geomagnetic stations of GNC through spline function, the contour of Δ Zm in China for each month in 2014 as an instance of high solar activity year is shown in Fig. 5. The Δ Zm in

mid-high latitude regions is much less than it in the low latitude regions. The difference between these two regions could change from about 10 to 30 nT in different months. The maximum of Δ Zm appears in August with the value about 55.1 nT, and the minimum of Δ Zm is about 5.7 nT emerges in January. This picture also indicates the differences of variation in daily variation amplitude of Z component for different seasons, seen as March, June, September and December representing spring, summer, autumn

and winter. The Δ Zm is largest in summer, and least in winter.

Fig. 6 expresses the contour of Δ Zm in China for each month in 2019 as an example of low solar activity year. Similar to the Δ Zm in 2014, the Δ Zm in mid-high latitude regions is less than it in the low latitude regions. The difference between these two regions could also change from about 10 to 30 nT in different months. The maximum of Δ Zm appears in June with the value about 44.6 nT, and the minimum of

Δ Zm is about 3.8 nT emerges in December. The seasonal change of Δ Zm is same to that in 2014 (high solar activity year), but the amplitude is much less than it in 2014 for each month in entire.

As the method mentioned in section 2, the Lloyd's season is suitable for analysis the seasonal variation of geomagnetic field. Fig. 7 shows the seasonal variation of Δ Z (Δ Zs) in 2014 and 2019. The black line indicates 2014, while the blue line indicates 2019. The data from all the stations of GNC in specific four

months are averaged to represent the daily variation amplitude of component Z in China for each season, on geomagnetic quiet time. In equinox months (E), the Δ Zs in 2014 is about 28.3 nT, while it is 23.2 nT in 2019. In summer solstice months (J), the Δ Zs in 2014 and 2019 are about 31.4 nT and 23.4 nT respectively. In winter solstice months (D), the Δ Zs in 2014 is about 20.5 nT and it is about 13.6 nT in 2019. The result indicates that whatever in the high or low solar activity year the value of Δ Zs are the

highest at the summer solstice, the second at the equinox and the lowest at the winter solstice. Fig. 7 also

expresses the seasonal change △ Zs is higher in high solar activity year than it in low solar activity year

with the average difference from 4.9 nT to 8.0 nT .

**4 The seasonal variation △ Zs in twelve years**

According to the Lloyd's season, the variation △ Zs in different seasons during the twelve years from

2008 to 2019 is analyzed. Fig. 8 shows the variation △ Zs of equinox months for every year. During

equinox months, the △ Zs in north China is much less than it in south China. The difference between

them could be above 20 nT. In some high solar activity years, the maximum of △ Zm contour is above 40

nT, as the year from 2010 to 2015. And there are two peaks with the value 47.2 nT and 46.7 nT exist in

2011 and 2014 respectively. In low solar activity years, the maximum of △ Zs contour for equinox

months each year is about 30 nT. It is smaller than it in high solar activity years. Whether in high or in

low solar activity years, the maximum of △ Zs contour almost appears in Yunnan province located in

southwest China.

Similar to Fig. 8, Fig. 9 indicates the variation △ Zs of summer solstice months for every year.

Comparison with the equinox months each year, the variation △ Zs of summer solstice months is much

higher at most time. But in some high solar activity years like 2011 and 2012, the variations of △ Zs are

very close between summer solstice months and equinox months . Also, the difference of △ Zs between

north China and south China is similar to that in equinox months with the value about 20 nT. In

high solar activity years, the maximum of △ Zs contour is even above 50 nT, for examples in the year

2014 and 2015. Even in low solar activity years, the maximum of △ Zs contour could reach to nearly 40

nT. Similar to the equinox months, the maximum of △ Zs contour almost appears in Yunnan province

both in high solar activity year and in low solar activity year.

The contour of △Zs in China for winter solstice months is expressed by Fig. 10. It indicates that the △Zs

in winter solstice months is much smaller than it in equinox months and summer solstice months. In most

of years, the difference of △Zs between north China and south China is less than 20 nT in winter solstice

months. The maximum of △Zs contour is about 35.5 nT appears in 2014 a high solar activity year.

Nevertheless the maximum of △Zs contour is less than 30 nT in other years. Also, the maximum of △Zs

contour in winter solstice months appears in Yunnan province both in high solar activity year and in

low solar activity year. This feature will be analyzed in detail in the section 5.

In order to study the overall state of variation △ Zs in China, the average of △ Zs from all the stations of

GNC is calculated for each Lloyd's season during the twelve years. Fig. 11 (a) ~(c) show the variation

△ Zs for the equinox, summer solstice and winter solstice months respectively in twelve years. It is very

evident to indicate the variation of △ Zs changing with the solar activity. All the three seasons have the

bigger value of △ Zs in high solar activity years (2010 to 2016). The variation of △ Zs for summer

solstice months is larger than it for the other two season months at most of time. Similar to △ Zs shown in

Fig. 8 and Fig. 9, the average of △ Zs for China in Fig. 11 is very close in equinox and summer solstice

months of 2011 and 2012 with the value 28.2 nT (2011 E) vs.29.4 nT (2011 J) and 28.2 (2012 E) vs.27.4

nT (2012 J). Especially in 2012, the △ Zs in equinox months is bigger than it in summer solstice months.

This phenomenon should be arisen from the factors which control the diurnal variation current system Sq,

like the conductivity and neutral wind in ionosphere, and will be discussed in section 6. The minimum

and maximum of △ Zs for equinox months are 20.2 nT in 2009 and 28.3 nT in 2015. For summer solstice

months, the minimum and maximum of △ Zs are 23.3 nT in 2019 and 31.6 nT in 2015. And the minimum

and maximum of △ Zs for winter solstice months are 12.8 nT in 2018 and 20.2 nT in 2015. The results

indicate that the maximum of △ Zs for winter solstice months almost equal to the minimum of △ Zs for

equinox months. These prove once again that the maximum and minimum of △ Zs appear in high and

low solar activity years for each season.

**5 Comparisons between stations in Yunnan and southeast regions**

As shown in the previous analysis, the maximum of ΔZm (or ΔZs) contour appears in Yunnan province

most of time both in high and low solar activity years, especially during equinox and summer solstice

months. In this section, three stations in Yunnan province will be analyzed further. As contrast, three

stations with similar latitude in southeast China are selected for study. Fig. 12 expresses the stations

distribution. YSH (Yongsheng), CHX (Chuxiong) and THJ (Tonghai) are stations in Yunnan province.

And SHW (Shaowu), QZH (Quanzhou) and XFJ (Xinfengjiang) are stations in southeast China. SHW

and QZH are located in Fujian province; XFJ is located in Guangdong province. Table 1 shows the

stations location information.

Similar to the method in section 4, the △ Zs of all the six stations are divided into Lloyd's three seasons.
For each Lloyd's season, three pairs of stations which are located in the similar latitude are made
comparison to analyse the △ Zs during the twelve years from 2008 to 2019. Three pairs are YSH vs.



SHW (pair 1); CHX vs. QZH (pair 2) and THJ vs. XFJ (pair 3). Also, the past twelve years are divided
into the high solar activity years (2010 to 2016) and low solar activity years (2008 to 2009 and 2017 to
2019).

The comparison is shown in Fig. 13. It is evident that the △ Zs of all the six stations have larger value in
high solar activity years than that in low activity years, whatever the season is. Fig. 13 (a) ~ (c) indicate
the equinox months. In pair 1, the △ Zs of YSH is larger than that of SHW with the average error 5.5 nT
in high solar activity years and 3.61 nT in low solar activity years. In pair 2, the average errors of △ Zs
between CHX and QZH are 1.7 nT and 1.9 nT in high and low solar activity years respectively. These
two stations shows very similar variation of △ Zs in twelve years. And the errors of them indicate no
obvious distinction between high and low solar activity years. The pair 3 of THJ and XFJ is in the lower
latitude. The average errors of △ Zs are much larger than that of the other two pairs, with 9.2 nT and 8.7
nT in high and low solar activity years respectively. Table 2 shows the average errors of △ Zs in three
pairs. High and Low in this table mean the high and low solar activity years. Based on the Fig. 13 and
table 2, among the three stations in Yunnan province, YSH and THJ located in north and south have
larger △ Zs than the stations with similar latitude in southeast China; CHX have similar △ Zs with
southeast station QZH. During the equinox months of twelve years, the average △ Zs of station THJ is
about 38.3 nT; YSH is second with the value of 35.2 nT; CHX is third with the value of 30.0 nT. This
result makes the contour of △ Zs in China have two extreme values in Yunnan province most of time
during the equinox months. Also, the maximum of the △ Zs contour usually appears in Yunnan province
for equinox months as shown in Fig. 8.

In summer solstice months, indicated by Fig. 13 (d) ~ (f), three pairs of stations indicate similar results
with the △ Zs variation in equinox months. The average errors of all the three pairs are more or less
higher than them in equinox months during high and low solar activity years (see table 2). In pair 1, the
difference of average errors between YSH and SHW are not big in high and low solar activity years with
the value about 6 nT. It is not obvious different for the △ Zs variation in Pair 2. The average error
between CHX and QZH is less than 3nT. But the most special pair is pair3. The average errors between
THJ and XFJ are very high with the value 12.2 nT and 8.4 nT in high and low solar activity years
respectively. This means that THJ and XFJ which are in the similar latitude express obvious different
geomagnetic diurnal variation in summer solstice months, particularly in high solar activity years. Also,
in summer solstice months during twelve years, the average △ Zs of station THJ is about 41.2 nT; YSH is
second with the value of 36.9 nT; CHX is third with the value of 31.0 nT. The △ Zs maximum of THJ
appears in 2015 with the value 53.2 nT. It is the biggest value for all the stations of GNC in Lloyd's
seasons during the recent twelve years on solar quiet day. It is also interesting that the △ Zs for each
station itself show similar values in equinox and summer solstice months during some high solar activity

years. For example, the Δ Zs of YSH has a difference less than 1 nT between equinox and summer

solstice months from 2011 to 2014 (high solar activity years). The similar result could also be seen from

other five stations.

Different from equinox and summer solstice months, it is notable that all the three pairs of stations in

winter solstice months don't express obvious difference. The Δ Zs variations of each pair are highly

consistent. It could be seen from table 2 that the average errors of Δ Zs in three pairs are about 2 nT.

These results indicate that the Δ Zs variations of the three stations in Yunnan province are very similar

with the stations at near latitude in southeast China. Moreover, the Δ Zs variations of all the six stations

are alike in winter solstice months, due to they are all in the similar latitude.

According to the analysis above, it is evident that the Δ Zs variations of three stations in Yunnan

province show different behaviors in equinox and summer solstice months. But in winter solstice months,

the Δ Zs variations of them are nearly the same. The comparisons between the stations in Yunnan

province and southeast China also confirm the distinction of the Δ Zs variations in YSH, CHX and THJ.

The solar activity is an important factor to affect the state of ionosphere. And the conductivity and neutral

wind in ionosphere could control the Sq current (Zhao et al., 2014). So the different Δ Zs variations of

stations in different seasons or in high and low solar activity years is preliminarily caused by the solar.

However, the discrepancy of Δ Zs variations among Yunnan stations is not only related to the solar

activity but also related to the other factor, may be the complicated electrical structures underground in

the local area. The combined action of solar and electrical structures underground makes the Δ Zs

variations in Yunnan stations much different in equinox and summer solstice months during high and low

solar activity years.

**6 Discussion and Conclusions**

In this paper, the daily variation amplitude of geomagnetic component Z on solar quiet days is analyzed

to study the geomagnetic diurnal variations in high and low solar activity years, in different seasons and

in different stations at similar latitude.

Based on the sunspot , the Δ Zm variations of China show much higher value for each month in high

solar activity years than it in low solar activity years. The geomagnetic daily variation on quiet days is

mainly caused by Sq current in ionosphere. The changes of conductivity and neutral wind in the





ionosphere are closely related to solar, and they mainly control the intensity and structure distribution of Sq current system (Pedatella et al., 2011). Also, the ionospheric conductivity makes notable change with solar activity, and increases significantly in the high solar activity year (Ji et al., 2006). So the ionosphere conductivity has a high correlation with the solar activity. Due to the ionospheric conductivity caused by

the electromagnetic radiation of the sun, it depends on the density and temperature of the neutral and ionized particles of the ionosphere (Xu 2003; 2009). As a result, the ionospheric conductivity directly makes the $\triangle$ Zm variations of China higher in high solar activity years for each month.

According to the Lloyd's season, the seasonal variations of the geomagnetic component Z daily variation amplitude show that the $\triangle$ Zs has bigger value in high solar activity years (2010 to 2016) for each season.

This is similar to the $\triangle$ Zm variations for each month in twelve years. The high correlation between the $\triangle$ Zs seasonal variations and solar activity years indicates the ionospheric conductivity affects the $\triangle$ Zs for each season in different years. The comparisons of $\triangle$ Zs variations in different seasons show that $\triangle$ Zs in summer solstice months is larger than it in equinox months at most of time. And the $\triangle$ Zs variations in winter solstice months is almost the minimal among the three seasons whatever in high or

low solar activity years. But during some high solar activity years, $\triangle$ Zs in equinox months is nearly equal to and even bigger than it in summer solstice months. The Sq current variations in the solar cycle studied by some researchers shows that the current intensity could reach maximum in equinox months (Campbell and Matsushita, 1982; Zhao, 2014). And the Sq current makes the geomagnetic component Z decrease near the noon in the north hemisphere of the Earth. So the geomagnetic component Z daily

variation amplitude could also reflect the size of the Sq current. Some researchers consider the Sq current reaching maximum in equinox months may be caused by the neutral wind in ionosphere (Amayenc, 1974; Matsushita and Xu, 1982). The neutral wind is mainly generated by the atmospheric tides which caused by the gravitational tide force and the thermal tide force of the sun (Richmond, 1989, 1995). The strength of the neutral wind in equinox months is higher than it in summer and winter solstice months during the

high solar activity years (Campbell and Matsushita, 1982; Yamazaki et al., 2009). The conductivity and neutral wind in ionosphere are two major factors affect the Sq current. The conductivity is highly consistent with the solar activity. So component Z daily variation amplitude of the same season (month) has bigger value in high solar activity years mainly caused by the conductivity in ionosphere. But the $\triangle$ Zs with high value in equinox months during some high solar activity years indicate that besides the

conductivity the neutral wind in ionosphere also makes great contribution to the Sq current. Based on the

discussions above, we could induce that the conductivity in ionosphere plays a major role in the seasonal variations of Sq current at most of time. When the neutral wind in ionosphere grows stronger and makes equal or greater effect than the conductivity, the Sq current intensity will become higher in equinox months than in other seasons during some high solar activity years.

The diurnal variations of geomagnetic field recorded by the ground stations on solar quiet days include the external field directly caused by Sq current and the internal field caused by the induced current underground along with the Sq current (Titheridge, 1995). The Sq current flowing anticlockwise makes the diurnal variations of Z component decrease rapidly near noon in north hemisphere (Campbell, 1997). The focuses of the Sq current usually locate near 30 degree in latitude for north and south hemisphere

respectively (Xu, 2003). Due to the structure of Sq current and the locations of the stations of GNC, the daily variations of Z show the value in the south China is larger than that in the north China. It is worthy of note that the maximum of Z diurnal variations amplitude always appears in Yunnan province. The diurnal variations of geomagnetic field caused by Sq current normally perform in latitude and local time, and the internal field is usually about half of the external field in strength (Xu, 2009). Comparisons

between stations in Yunnan province and southeast China show that except CHX, the Z diurnal variations amplitude of YSH and THJ indicate much larger than it of the stations in the similar latitude in southeast China during equinox months and summer solstice months. These results maybe show that besides the influence from conductivity and neutral wind in ionosphere as discussed above, the complicated underground electrical structures in Yunnan province could also make great effect to the

internal field caused by the induced current underground. In China, the abnormities of the conductivity underground were discovered along the Coast of Bohai Sea, the eastern part of Gansu Province, the area around Tangshan and Yunnan Province (Chen, 1974; Xu et al., 1978). The abnormities of the conductivity underground not only relate to the composition and temperature of rocks, but also relate to some geophysics phenomenon like the seismic wave low velocity area and geothermal flow anomaly

area (Rikitake, 1966). Yunnan Province is located in a special geographical location which has strong tectonic activity (Yuan et al., 2015). Hou and Shi (1984) using the variations of Z component to analyze the Wiese vector distribution during storm sudden commencement (SSC) and bay disturbance events infer that there are some high conductivity underground in local regions of Yunnan Province. The geomagnetic variation abnormalities perform evidently in the variation of Z component (Xu et al., 1978;

Li et al., 2006). If the variations of Z component have significant differences between stations in near


local region, there is transverse inhomogeneity of conductivity underground in this region (Rikitake, 1966; Tang et al., 1999). The performances of YSH and THJ in the daily variations of Z different from CHX indicate the conductivity underground of these two stations is higher than it of CHX. Moreover, the conductivity underground of THJ is higher than it of YSH. If we set the stations at similar latitude in the

southeast China as reference background stations, the difference of Z diurnal variations amplitude between the stations in Yunnan Province and reference stations could be considered as the enhanced induced field (internal field) caused by the higher conductivity underground. According to the comparison, the enhanced induced field could increase the Z diurnal variations amplitude of YSH from12.8% to 18.6%. For THJ, the contribution from the higher conductivity underground could make

the enhanced induced field be about 30% of the Z diurnal variations amplitude in equinox months and reach above 40% of the Z diurnal variations amplitude in summer solstice months during high solar activity years. Table 3 shows the percentage of the enhanced induced field in the Z diurnal variations amplitude. High and Low in this table mean the high and low solar activity years. The high conductivity underground leads to the internal field larger than half of the external field in strength. So the high

conductivity underground likes an amplifier on the basis of the conductivity and neutral wind in ionosphere to affect the geomagnetic diurnal variations. When the external field increases, the effect of the amplifier from the high conductivity underground is more evident. However, in the winter solstice months, due to the effects from the conductivity and neutral wind in ionosphere are weaker than in other seasons, the amplifier contribution from the conductivity underground to the geomagnetic diurnal

variations is also small. These could be seen from the comparisons between the three stations in Yunnan Province and the stations in the similar latitude in southeast China during winter solstice months.

This paper analyses the daily variation amplitude of geomagnetic component Z from GNC on solar quiet days in the past twelve years, the following understanding is obtained:

(1) The ionospheric conductivity mainly contributes to the change of Z daily variations amplitude along

with solar activity in the same month or season.

(2) The neutral wind in ionosphere plays an important role in making the strength of Z daily variations amplitude higher in equinox months during some solar high activity years.

(3) The conductivity underground acts as an amplifier to make the Z daily variations amplitude in Yunnan Province distinct with it in other area especially in equinox and summer solstice months.

Due to the daily variation amplitude of geomagnetic component Z is an important data product of GNC, this paper based on it shows the overall characteristics of Z daily variation amplitude since the GNC digitization in 2008, makes some interpretations for it, and tries to provide the reference background for researchers who use the geomagnetic component Z daily variation amplitude to analyze the anomaly. The future work will be further studied using more stations constructed, especially in the West China

based on the GNC development plan for next decade. Also, the combination with the model of the conductivity underground such as MT is needed to improve our interpretations.

**Acknowledgements.**

This research was supported by the National Key R&D Program of China (Grant no. 2018YFC1503504), National Natural Science Foundation of China (41774085), the Special Fund of the Institute of

Geophysics, China Earthquake Administration (Grant Number DQJB20B26), and the Strategic Priority Research Program of Chinese Academy of Sciences (Grant No. XDA14040403, XDA14040404). The magnetic field data of the ground stations is from the Geomagnetic Network of China. The sunspot number data is from Sunspot Index and Long-term Solar Observations (SILSO data/image, Royal Observatory of Belgium, Brussels). The list of international geomagnetic quiet days is from the World

Data Center for Geomagnetism, Kyoto, Japan.

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

**Table1. The compared stations location.**

| Stations in Yunnan | | | Stations in southeast China | | |
|---|---|---|---|---|---|
| Code | Longitude | Latitude | Code | Longitude | Latitude |
| YSH | 100.7 ˚E | 26.7 ˚N | SHW | 117.4 ˚E | 27.3 ˚N |
| CHX | 101.5 ˚E | 25.0 ˚N | QZH | 118.3 ˚E | 24.9 ˚N |
| THJ | 102.7 ˚E | 24.1 ˚N | XFJ | 114.6 ˚E | 23.6 ˚N |

**Table2. The average errors of Δ Zs in three pairs of stations.**

| | Paris | Equinox months(E) | | Summer solstice months (J) | | Winter solstice months (D) | |
|---|---|---|---|---|---|---|---|
| | | High | Low | High | Low | High | Low |
| 1 | YSH and SHW | 5.5 nT | 3.6 nT | 6.3 nT | 5.2 nT | 2.2 nT | 1.2 nT |
| 2 | CHX and QZH | 1.7 nT | 1.9 nT | 2.6 nT | 2.7 nT | 2.1 nT | 1.9 nT |
| 3 | THJ and XFJ | 9.2 nT | 8.7 nT | 12.2 nT | 8.4 nT | 2.4 nT | 1.9 nT |

**Table3. The percentage of enhanced induced field in the Z diurnal variations amplitude.**

| | Paris | Equinox months(E) | | Summer solstice months (J) | |
|---|---|---|---|---|---|
| | | High | Low | High | Low |
| 1 | YON and SHW (refer) | 16.6% | 12.8% | 18.6% | 18.2% |
| 2 | THJ and XFJ (refer) | 28.2% | 33.8% | 41.2% | 29.3% |





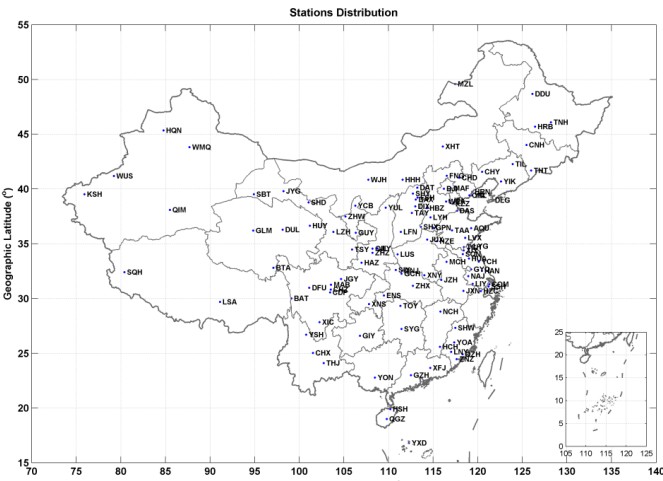

**Figure 1.  The distribution of stations in GNC.**


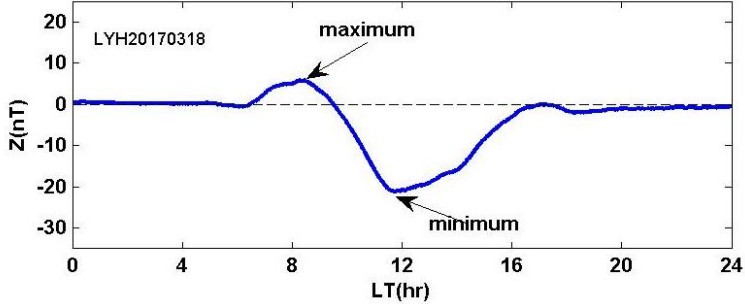

**Figure 2. The calculation of component Z daily variation amplitude.**





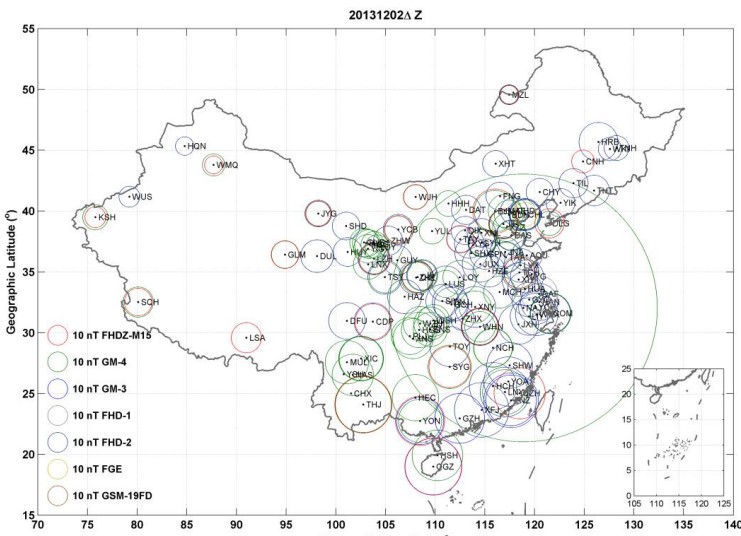

**Figure 3. The confirmation of Z component daily variation amplitude. The different colour circles represent the different instruments in stations. The circles in the left bottom indicate the size corresponding to 10 nT.**

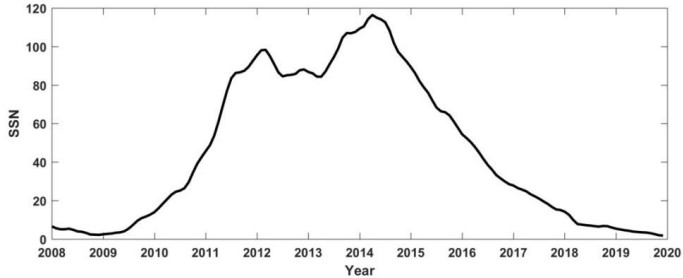

**Figure 4. The monthly mean total sunspot number variation from 2008 to 2019.**


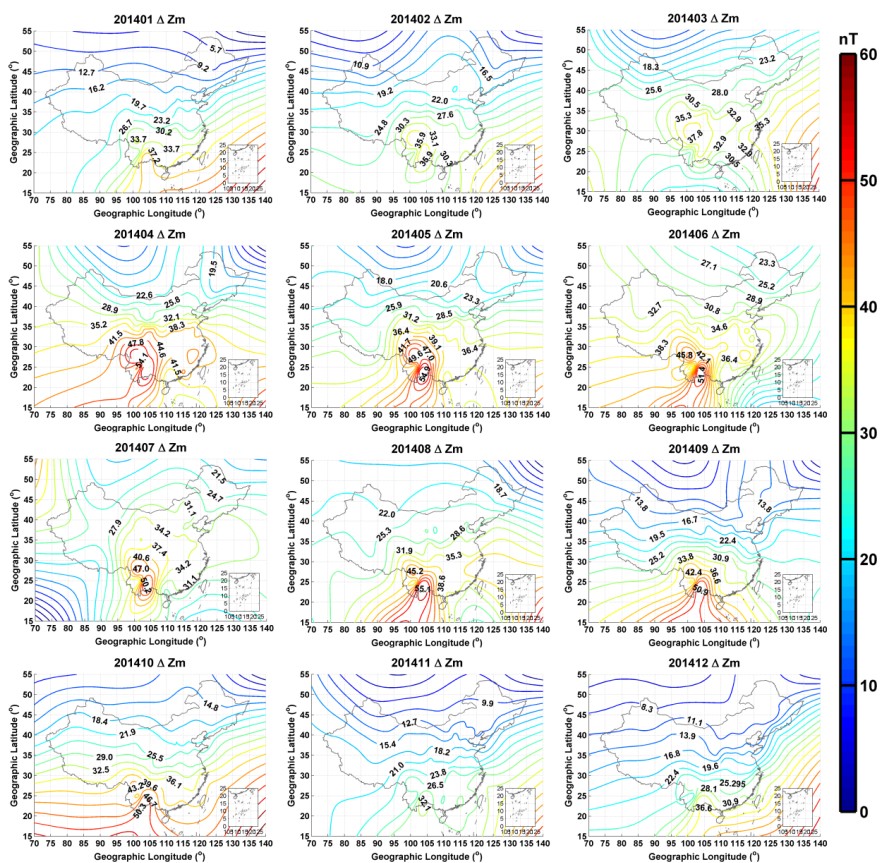

**Figure 5. The contour of monthly variation Δ Zm in China for each month in 2014.**



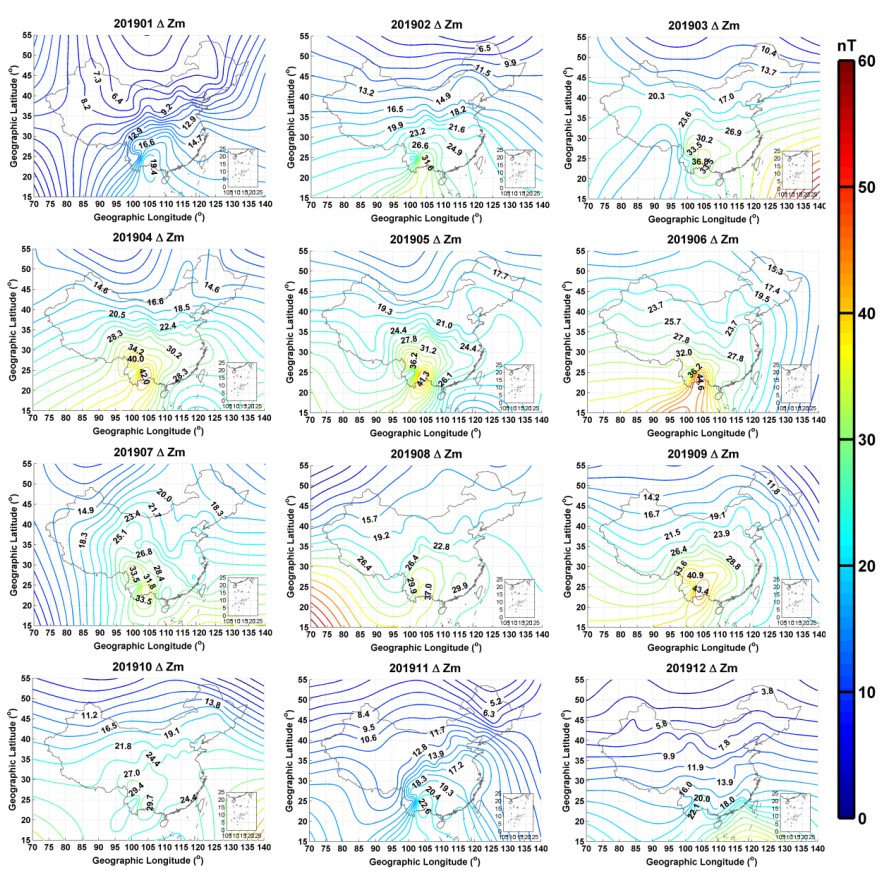

**Figure 6. The contour of monthly variation △ Zm in China for each month in 2019.**

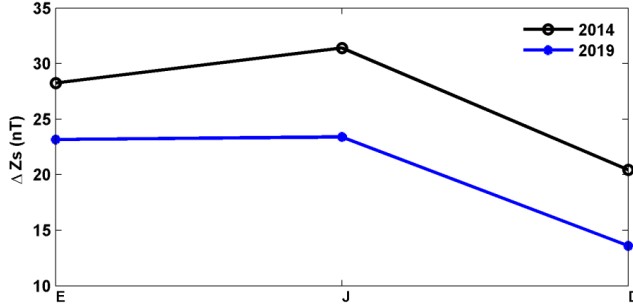

**Figure 7. The averaged seasonal variations of △ Zs for GNC in 2014 and 2019. The black and blue lines represent the variations in 2014 and 2019 respectively at the equinox, summer solstice and winter solstice months.**


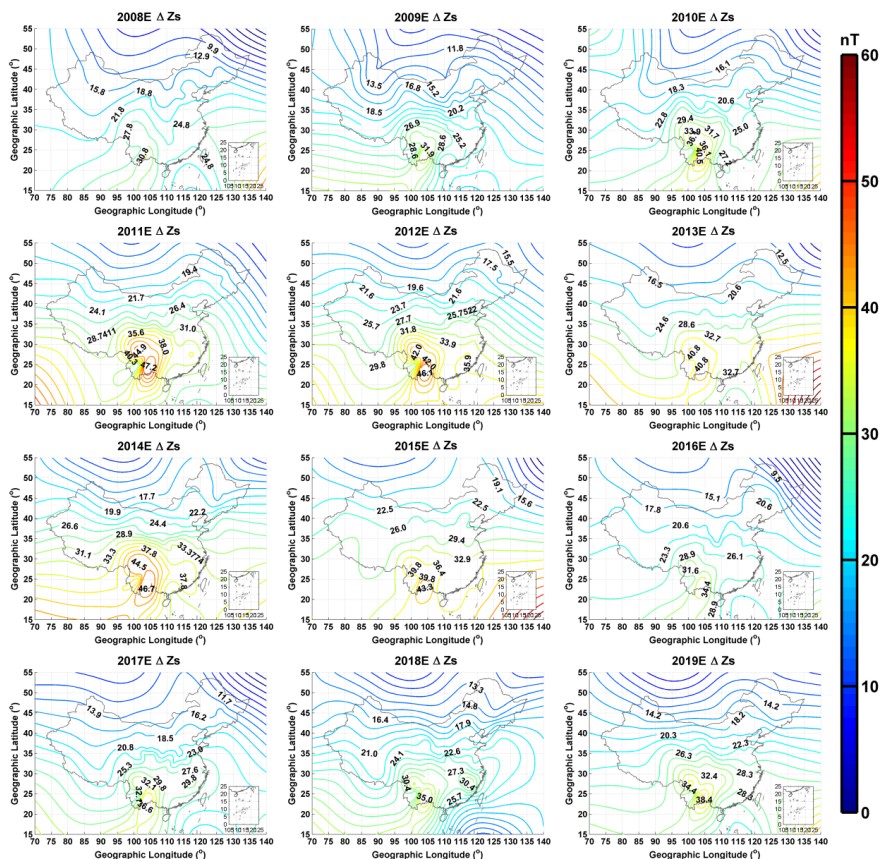

**Figure 8. The contour of seasonal variation ∆ Zs in China for equinox months in twelve years.**

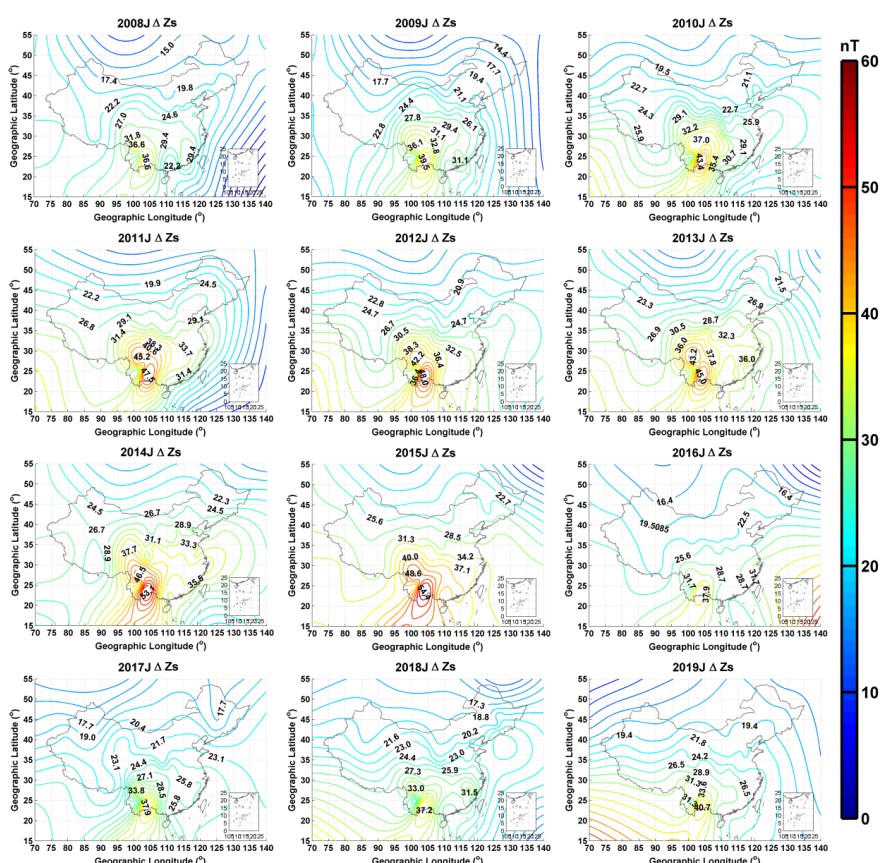

**Figure 9. The contour of seasonal variation ∆ Zs in China for summer solstice months in twelve years.**



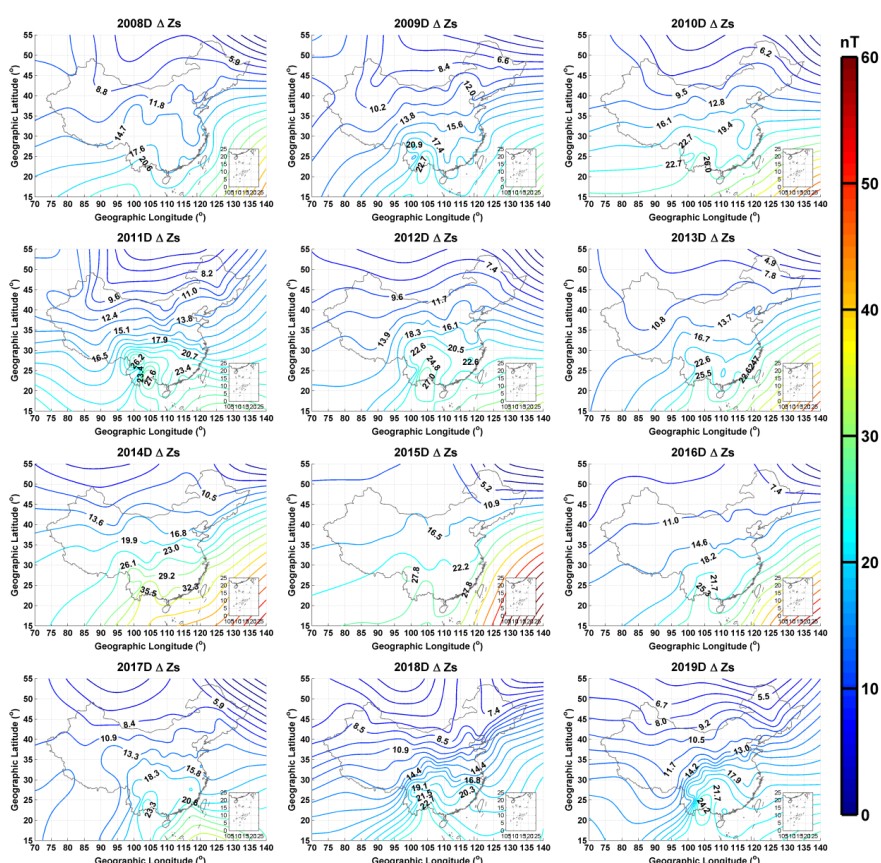

Figure10. The contour of seasonal variation Δ Zs in China for winter solstice months in twelve years.





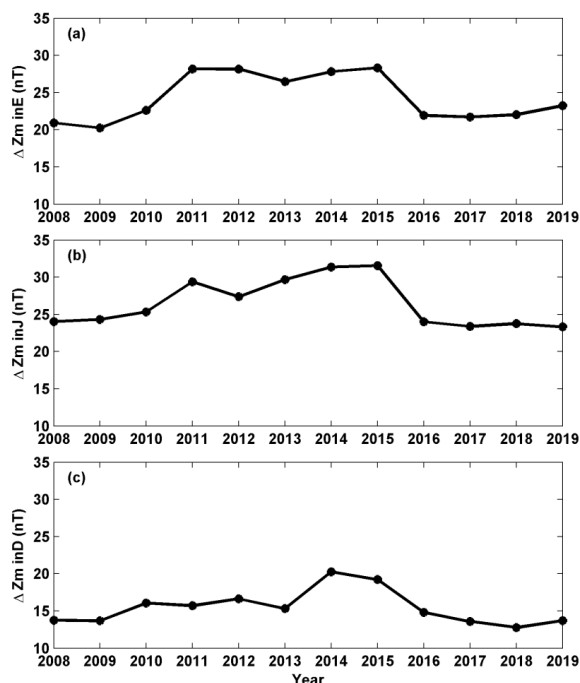

Figure11. The averaged seasonal variations of Δ Zs for GNC from 2008 to 2019. (a) ~ (c) in the figure
        represent the equinox, summer solstice and winter solstice months.

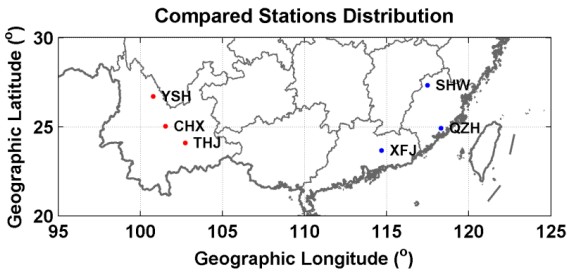

Figure 12. The compared stations distribution. The red dots represent the stations in Yunnan Province and
the blue dots represent the stations in southeast China.



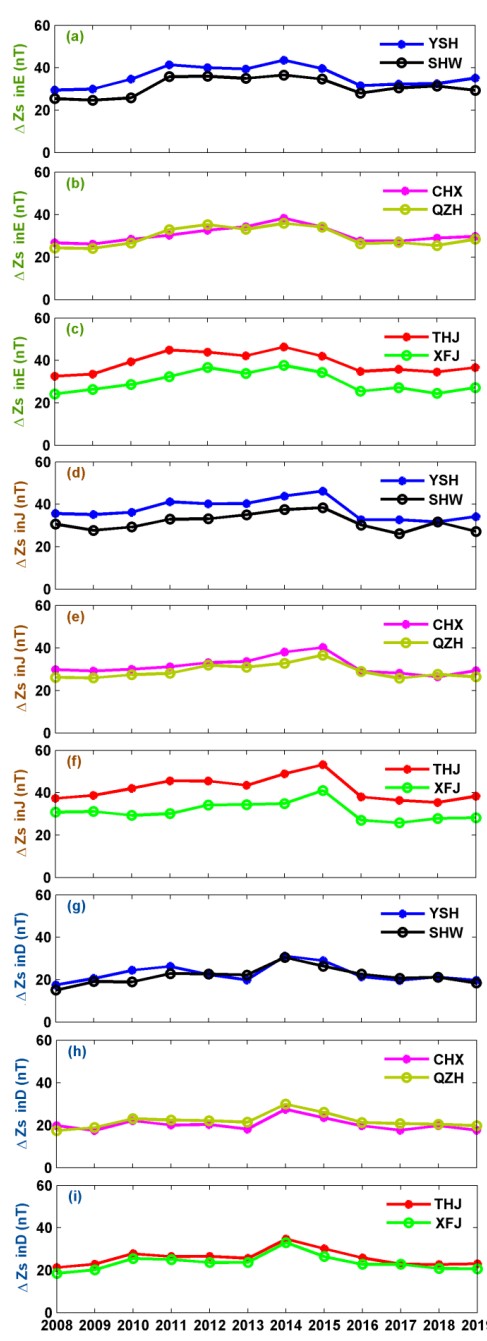

**Figure 13. The comparison of Δ Zs for three pairs of stations in different seasons. The different color lines represent different stations. The seasons are (a) ~ (c) for the equinox months; (d) ~ (f) for the summer solstice months; (g) ~ (f) for the winter solstice months.**