# Peer review of "Analysis of the Geomagnetic Component Z Daily Variation Amplitude Based on the China Geomagnetic Network"

_Geoscientific Instrumentation, Methods and Data Systems, 2020_

## Short Comment (SC1) · 29 Dec 2020

This article uses twelve years data from the Geomagnetic Network of China to analyse the daily variation amplitude of Z. The idea of this manuscript is original. And it may provide some background information to researchers who use the daily variation amplitude of geomagnetic component Z. The comments below may be helpful to improve the quality for publication. 1. Figure 13 maybe need to be reshaped, for example, each column represents every seasonal months and each row represents every comparison pair of stations. This distribution of figure could make the comparison more evident. 2. Line 270 which mentioned "The changes of conductivity and neutral wind in the ionosphere are closely related to solar, and they mainly control the intensity and structure distribution of Sq current system." should also include the contribution of the geomagnetic field. The geomagnetic field interacts with neutral wind producing the electric field of ionospheric generator. The geomagnetic field maybe need to add some discussion to illustrate whether it has effect in this study. The manuscript is well written and the results are reasonable. I would hope this article will be published in this journal.

———————————————

---

## Author Comment (AC1) · 30 Dec 2020

Thank you for your kind comments. 1. The author will rejust Figure 13 in 3 rows and 3 columns to show the comparison more clear. 2. The geomagnetic field from the earth interacts with neutral wind in the ionosphere producing the electric field of ionospheric generator. The combination of this electric field and the conductivity in the ionosphere makes effects to the Sq current system. The geomagnetic field is mainly from the main field of the earth. The period of the main field could range from about thirty years to ten thousand years. It has little change in about twelve years. On the other hand, the main field in the local region which the stations distribute is almost the same. The effect of

the geomagnetic field could be ignored. And the conductivity and neutral wind in the ionosphere are the major facts influence the Sq current system. These supplements will be added in the updated manuscript.

---

## Short Comment (SC2) · 14 Jan 2021

The manuscript tried to analyze the feature of component Z daily variation amplitude based on the China Geomagnetic Network, such as the Z daily variation amplitude in different months, the seasonal variation, and comparison the Z daily variation amplitude of Yunnan province with other regions. The data used in this study has an important contribution to the analysis of the geomagnetic diurnal variation or to analyze the anomaly. The following are some suggestions to improve the manuscript. (1) Page 2, lines 56-61, this sentence is too long. Rewrite. (2)Page 7,line 204, delete this sentence "SHW and QZH are located in Fujian province; XFJ is located in Guangdong

province.".The reason is that the Fujian province and the Guangdong province are not marked in any figure, and it is not important.

---

## Author Comment (AC2) · 18 Jan 2021

Thank you for your good suggestions. 1. The sentence in lines 56-61, page 2 will be adjusted as the follows." We will just intensive study the feature of component Z daily variation amplitude in China using the data after the completion of digital network until now on solar quiet days (2008-2019). The Z daily variation amplitude in different months, seasonal variation during more than ten years and comparisons of the Z daily variation amplitude between Yunnan province (in southwest of China) with other regions are studied in this paper." 2. The sentence in line 204, page 7 will be deleted in the updated manuscript.

---

## Referee Comment (RC1) · Anonymous Referee #1 · 17 Feb 2021

This paper investigated the daily variation amplitude of geomagnetic component Z from GNC for about a whole solar cycle. Three useful results were obtained: The Z daily variations amplitude in the same season were due to the different ionospheric conductivity in different phases of the solar cycle; The neutral wind could be the main factor for the stronger effect of the Z daily variations amplitude in equinox months; The 12%-41% of the Z daily variations amplitude increase in Yunan Province could be attributed to the complicated underground electrical structures. In general, this paper is well organized and written and the results from this paper are important for the further study of the Sq current structures as well as the underground electrical conductivity. I recommend it to be published in this article.

---

## Author Comment (AC3) · 18 Feb 2021

Thank you very much for the referee's review and encourages. The authors will make further study about the daily variation caused by Sq current in future.

---

## Referee Comment (RC2) · Anonymous Referee #2 · 19 Feb 2021

General comments

The manuscript is devoted to systematic analysis of Z component amplitude during magnetically quiet days over the 24th solar cycle recorded at magnetic stations of the Chinese network.

One of the main outcomes declared is that the conductivity and neutral wind in iono-sphere controlling the Sq current eventually affect Z amplitude of the quiet daily varia-tion. It is clear that the geomagnetic daily variation on quiet days is mainly controlled by Sq current system and the authors are well aware of that (see section 6). However, in the present manuscript no study of specifically the conductivity and neutral wind is

presented, so the claimed result is questionable.

Second, why the authors chose exactly the Z amplitude for the study remains unclear. The horizontal components of the magnetic field just as much reflect the variability of the Sq current system.

Also, authors argue that induced terrestrial currents might systematically affect the quiet daily Z amplitude. However, no discussion on crustal magnetic field producing inhomogeneous static magnetic anomalies is given. The 'induced current' argument sounds doubtful as GICs are (1) short-period and (2) accompany strong disturbances at high latitudes, whereas the study involves only quiet days. Indeed, GICs are dependent on the crustal conductivity. Another possible explanation of the observed bias between stations may be simply due to different scale factors applied to vector magnetometers installed at magnetic stations under consideration. Otherwise, some comments are required with respect to data validation and correction procedures as well as instruments installed at compared stations (the presented quality control description is too general). Since the authors try to identify such tiny effects of the order of magnitude of 1+ nT, it should be confirmed that all the instruments under consideration are intercalibrated, short-period drifts are removed from data, etc.

Specific comments

Some logical contradictions have been detected in the text. For example, the authors state in the Introduction section that "Among the various quality control methods for geomagnetic data, the daily variation amplitude of geomagnetic component Z is an effective data product to analyze the data quality for various instruments in the same station or to compare the data quality with different stations nearby." However, in the course of the study they compare data between the stations to identify some regional geomagnetic effects, which excludes quality control.

The manuscript contains inaccurate and frankly erroneous statements. Herein, I list just several ones. Line 36 says that "The current vortex in the northern hemisphere

flows clockwise", however this statement is wrong. The Sq flow direction in the northern hemisphere is counterclockwise. Line 44 says that "The variable magnetic field could produce the induced magnetic field", however it induces currents. Line 328 says that "The geomagnetic variation abnormalities perform evidently in the variation of Z component." – in addition to poor English, variation apparently affects variation.

Line 39 says that "Based on this phenomenon, the daily variation amplitude of Z component could represent the strength of Sq current and could also be used for different instruments or stations to compare each other to confirm if the data quality is up to standard." The variability of the Sq current system is represented just as much in the horizontal (X and Y) geomagnetic components. See e.g. Soloviev, A., Smirnov, A., Gvishiani, A., Karapetyan, J., Simonyan, A. (2019), Quantification of Sq parameters in 2008 based on geomagnetic observatory data, Advances in Space Research, Vol. 64, Iss. 11, 2019, pp. 2305-2320, doi: https://doi.org/10.1016/j.asr.2019.08.038, where the authors study X, Y and Z daily amplitudes to quantify various Sq features and eventually reconstruct the equivalent current system. This paper is also worth mentioning as the authors use original method for accurate determination of magnetically quiet days for further analysis of Sq variations. Some discussion on the Earth's conductivity and geomagnetically induced currents affecting Z variations is also present there.

All mentioning of possible relation of geomagnetic variations with earthquakes should be removed along with the corresponding references. This topic is highly under debate and the manuscript does not provide any outcomes of such studies.

Line 48: what is "seismic geomagnetism"?

Line 61: "This study will provide the background information or reference to researchers who use the daily variation amplitude of geomagnetic component Z to study the geomagnetic diurnal variation or to analyze the anomaly." –Aren't "daily variation" and "diurnal variation" the same thing? What kind of anomaly is mentioned?

According to Fig.2, the Z baseline is exactly 0, which looks suspicious; no comments

are given in the text with respect to baseline determination.

Line 79-80: according to Sq schema the Z second extremum should be observed near local noon (where it's minimal), but not in the afternoon. This is by the way illustrated by Fig. 2.

Lines 85-94: is the described quality control procedure done manually or automated somehow? Clarification should be added.

Lines 95-101: in my opinion, using international quiet days to study fine regional geomagnetic features is incorrect. These days reflect the magnetic quietness in a planetary scale and admit some local insignificant disturbances.

Line 116 says that "According to the frequency and average number of sunspots on the surface of the sun, the solar activity has a period of about 11 years", however solar periodicity has much more deep justification including solar dynamo theory (see e.g. Charbonneau P. Solar Dynamo Theory // Annual Review of Astronomy and Astrophysics. 2014. Vol. 52(251). https://doi.org/10.1146/annurev-astro-081913-040012, Charbonneau P. Dynamo Models of the Solar Cycle // Living Rev. Sol. Phys. 2005. Vol.2(2), https://doi.org/10.12942/lrsp-2005-2).

Lines 118-120 and Figure 4 caption: to be more precise, the plot shows 13-month smoothed monthly sunspot number.

Figures 5,6,8,9,10: What interpolation method was used? I see the isolines outside China, which suggests that data from foreign observatories were also used in the processing.

Lines 126-140: These considerations along with Figs. 5-6 are mainly declarative and do not bring any new suggestions. For instance, no interpretation is given for "deltaZm in mid-high latitude regions is less than it in the low latitude regions". The fact that in local summer deltaZ is higher than in local winter is pretty obvious. What is more interesting is the analysis of the dynamics of the northern Sq vortex focus position over

the considered period, which falls on the south of China.

Line 146: it follows that over 2019 deltaZs values in E (23.2 nT) and J (23.4 nT) seasons are approximately the same. The argumentation given in the Discussion section does not explain the fact that over 2014 deltaZs values in E are lower than in J seasons. It only states that the neutral wind sometimes makes a comparable contribution to Z variations as the conductivity. Do the authors suggest that year 2019 was exactly the case? Is it possible to provide some data on conductivity and/or neutral wind for 2014 and 2019?

Line 152: What is "average difference"?

Lines 154-178: Again, this fragment is purely declarative. For example, how do authors interpret "there are two peaks with the value 47.2 nT and 46.7 nT exist in 2011 and 2014 respectively"? What kind of new information is provided by giving all 48 charts in Figs. 8-10?

Lines 212-229: what is meant by the "average error"? As stated further by the authors, the difference is attributed to different regional crustal conductivity, so it can't be treated as an error.

Lines 250-251: it says that "deltaZs variations of all the six stations are alike in winter solstice" as all six stations are "in the similar latitude". Taking into account further argumentation, this phrase does not explain the variation similarity in winter and should be removed. Otherwise clarification is required.

Lines 310-311: Taking into account the seasonal variability of the northern Sq loop focus, station locations in Fig. 3 and the fact that at the longitude under consideration the geomagnetic latitude of 30 degrees (corresponding to average focus position of the northern Sq loop) corresponds to 40N geographic latitude, the statement should be revised.

Lines 345-350: The authors argue that the crustal conductivity acts as an amplifier of

external field variations and in winter this contribution decreases. At the same time, according to Fig. 7 deltaZs in D season of 2014 is approximately the same as in E season of 2019. According to the authors' logic it should lead to notable difference between neighboring stations in D season of 2014, but it is not the case according to Fig. 13. Also, the amplification effect suggested by the authors should lead to a larger differences in active years (e.g., 2014), however Fig. 13 demonstrates quasi constant bias throughout the entire solar cycle. Thus, the amplification effect is doubtful.

Technical corrections

English is heavily stricken and should be seriously corrected. Some corrected excerpts are given below: - Line 55: component Z deserves attention - Line 70: components - Line 84: data selection is carried out as follows - Line 90: to validate - Line 217: two stations show - Line 255: variations at YSH, CHX and THJ - Line 258: solar activity years are preliminarily caused by the solar factor - Line 260: related to the other factor, presumably the complicated electrical - Line 268: Based on the sunspot numbers - (everywhere) conductivity underground -> crustal conductivity

The sentence on line 56 (and many others throughout the text) should be rewritten: "We will just intensive study the feature of component Z daily variation amplitude in China using the data after the completion of digital network until now on solar quiet days (2008-2019), including study the Z daily variation amplitude in different months, analysis the seasonal variation during more than ten years and comparison the Z daily variation amplitude of Yunnan province (in southwest of China) with other regions.".

Line 115: "The generation of Sq current system is closely related to the sun, so the activity of the solar will directly affect the current system" – should be rewritten. Line 125: the URL doesn't work.

Figure 11: Should be deltaZs in vertical axes.

Some text fragments are too verbose and should be shortened. For example, the

fragment between lines 246-250 contains three sentences identical in meaning: "all the three pairs of stations in winter solstice months don't express obvious difference", "variations of each pair are highly consistent", "variations of the three stations in Yunnan province are very similar with the stations at near latitude in southeast China".

Lines 305-307: no information on induced terrestrial currents is given in the mentioned reference (Titheridge, 1995).

Please avoid such references as Xu, W. Y.: Geomagnetism, Beijing: Seismological Press, 2003; Xu, W. Y.: Physics of Electromagnetic Phenomena of the Earth, Hefei: Press of University of Science and Technology of China, 2009; Rikitake, T.: Electromagnetism and the Earth's Interior, Amsterdam: Elsevier, 1966 and use references to particular chapters instead.

The text of the manuscript in its present form is raw and unacceptable. It should be rethought and rectified; the focus and main results of the study should be outlined more accurately and clear.